# Chemical Transport Model Simulations of Organic Aerosol in Southern California: Model Evaluation and Gasoline and Diesel Source Contributions

Shantanu H. Jathar[1], Matthew Woody[2], Havala O. T. Pye[2], Kirk R. Baker[2], and Allen L. Robinson[3]

[1] Mechanical Engineering, Colorado State University, Fort Collins CO 80525
[2] US Environmental Protection Agency, Research Triangle Park, NC, 27711
[3] Mechanical Engineering, Carnegie Mellon University, Pittsburgh, PA 15213

*Correspondence to*: Shantanu H. Jathar (Shantanu.Jathar@colostate.edu) or Allen L. Robinson (alr@andrew.cmu.edu)

**Abstract.** Gasoline- and diesel-fueled engines are ubiquitous sources of air pollution in urban environments. They emit both primary particulate matter and precursor gases that react to form secondary particulate matter in the atmosphere. In this work, we updated the organic aerosol module and organic emissions inventory of a three-dimensional chemical transport model, the Community Multiscale Air Quality Model (CMAQ), using recent, experimentally-derived inputs and parameterizations for mobile sources. The updated model included a revised volatile organic compound (VOC) speciation for mobile sources and secondary organic aerosol (SOA) formation from unspeciated intermediate volatility organic compounds (IVOC). The updated model was used to simulate air quality in southern California during May and June 2010 when the California Research at the Nexus of Air Quality and Climate Change (CalNex) study was conducted. Compared to the Traditional version of CMAQ, which is commonly used for regulatory applications, the updated model did not significantly alter the predicted organic aerosol (OA) mass concentrations but did substantially improve predictions of OA sources and composition (e.g., POA-SOA split), and ambient IVOC concentrations. The updated model, despite substantial differences in emissions and chemistry, performed similar to a recently released research version of CMAQ (Woody et al., 2016) that did not include the updated VOC and IVOC emissions and SOA data. Mobile sources were predicted to contribute 30-40% of the OA in southern California (half of which was SOA), making mobile sources the single largest source contributor to OA in southern California. The remainder of the OA was attributed to non-mobile anthropogenic sources (e.g., cooking, biomass burning) with biogenic sources contributing to less than 5% to the total OA. Gasoline sources were predicted to contribute about thirteen times more OA than diesel sources; this difference was driven by differences in SOA production. Model predictions highlighted the need to better constrain multi-generational oxidation reactions in chemical transport models.

# 1 Introduction

Organic aerosol (OA) is a major component of atmospheric fine particulate matter (Jimenez et al., 2009). Source apportionment studies have historically attributed the majority of ambient OA in Southern California to motor vehicle emissions (Schauer et al., 1996), but analysis of data from the California Research at the

Nexus of Air Quality and Climate Change (CalNex) study has led to conflicting conclusions about the overall contribution of motor vehicles to OA in southern California and the relative importance of gasoline versus diesel sources. Bahreini et al. (2012) hypothesized that the majority of OA in southern California was secondary organic aerosol (SOA) formed from emissions from gasoline powered-sources based on differences in weekday and weekend pollutant concentrations; Hayes et al. (2013) and Zotter et al. (2014)

reached the same conclusion based on analysis of mass spectrometer and radiocarbon data respectively. In contrast, Gentner et al. (2012) concluded that diesel vehicles contributed more OA than gasoline vehicles based on a comprehensive speciation of SOA precursors present in gasoline and diesel fuels. Ensberg et al. (2014) proposed that observed levels of OA could be explained only if vehicle emissions were a minor source of SOA or that the SOA formation potential of vehicle emissions was significantly higher than that measured

in laboratory studies. Finally, source-resolved chemical transport model (CTM) simulations predicted that gasoline sources contributed approximately twice as much POA as diesel sources in southern California (Woody et al., 2016).

Research and regulatory efforts have historically focused on emissions of primary organic aerosol (POA) but

recently the attention has shifted to secondary organic aerosol (SOA) since SOA appears to dominate OA mass concentrations even in urban areas (Zhang et al., 2007). Typical CTM treatments of OA assume non-volatile POA emissions and formation of SOA from "traditional" precursors (Carlton et al., 2010), which are speciated volatile organic compounds (VOCs) such as alkanes smaller than $C_{12}$, single-ring aromatics, isoprene and mono- and sesquiterpenes. Robinson et al. (2007) proposed a new conceptual model for

emissions and evolution of OA from combustion sources : (1) POA emissions are semi-volatile and reactive (Grieshop et al., 2009;Huffman et al., 2009;May et al., 2013a;May et al., 2013b, c), (2) combustion sources emit substantial amounts of intermediate volatility organic compounds (IVOCs) that are efficient SOA precursors (Jathar et al., 2014;Zhao et al., 2015) and (3) semi-volatile organic vapors in equilibrium with OA photo-chemically react or "age" in the atmosphere to form additional SOA (Miracolo et al., 2010). Recent

state-of-the-science OA models have included these three processes, which have improved model performance (Murphy and Pandis, 2009;Koo et al., 2014). These improvements, however, have required simultaneous inclusion of all the above-mentioned processes; for example, inclusion of semi-volatile POA without SOA formation from IVOCs and aging reactions degraded model performance vis-à-vis total OA

mass (Robinson et al., 2007). However, the inputs required to represent these three processes are poorly constrained. For example, IVOC emissions from all sources are often assumed to be 1.5 times the POA emissions (Robinson et al., 2007;Shrivastava et al., 2008;Koo et al., 2014;Woody et al., 2016), based on measurements from two medium duty diesel vehicles (Schauer et al., 1999). New experimental data are

needed to better constrain these processes.

Recently, a series of experiments investigated the VOC emissions and SOA formation from gasoline vehicles, diesel vehicles, and small off-road engines recruited from the California in-use fleet (Gordon et al., 2014a;Gordon et al., 2014b;Gordon et al., 2013). May et al. (2014) analyzed the VOC data to develop detailed

emissions profiles. Jathar et al. (2014) analyzed the SOA data to derive quantitative estimates of the IVOC emissions and their potential to form SOA after several hours of atmospheric oxidation. Here, we use the term IVOCs to represent higher carbon number species ($C_{12+}$) that are difficult to speciate using traditional gas-chromatography mass-spectrometry techniques due to the very large number of constitutional isomers and/or polarity of partially oxidized species (Jathar et al., 2014;Presto et al., 2011;Zhao et al., 2015;Hatch et

al., 2015); Jathar et al. (2014) referred to these as unspeciated organic compounds. We use the term VOCs to include the class of SOA precursors typically speciated using conventional GC-MS techniques (e.g., alkanes smaller than $C_{12}$ and single-ring aromatics). Jathar et al. (2014) derived separate parameterizations to account for SOA formation from IVOC emissions from gasoline and diesel sources for use in CTMs.

In this work, we used an updated version of CMAQ to simulate ambient OA from gasoline and diesel sources in southern California. The updates included new mobile source emissions profiles for VOCs (based on May et al. (2014)) and emissions and parameterizations for SOA production from IVOCs (based on Jathar et al. (2014)). Model predictions were evaluated using data collected during CalNex, compared to predictions of other models, and used to investigate the contribution of gasoline and diesel sources to ambient OA

concentrations. This was the first time that a comprehensive set of gasoline and diesel source data have been used to develop source-specific IVOC inputs for a three-dimensional CTM. Earlier modeling efforts have relied on data that are almost a decade old (e.g., Koo et al. (2014)) and/or have used box models that may not accurately simulate horizontal and vertical transport and deposition (e.g., Hayes et al. (2015)). Hence, our work presents a step forward in improving the representation of sources, emissions and photochemical

production of OA in large-scale models. This paper builds upon recent work by Baker et al. (2015) and Woody et al. (2016) who used different versions of CMAQ to simulate OA in California during May and June 2010.

## 2 Methods

In this section, we provide a brief overview of CMAQ followed by more detailed descriptions of the OA model and emissions.

### 2.1 Chemical Transport Model

The CMAQ model version 5.0.2 was used to simulate air quality in California from May 4 to June 30, 2010, which coincides with the CalNex campaign (May–July 2010). Details about the application of this version to CalNex can be found in Baker et al. (2015) and Woody et al. (2016). Briefly, the model domain covered California and Nevada with a 4-km (317 x 236) grid resolution (Figure S.1). The vertical domain included 34 layers and extended to 50 mbar. Atmospheric gas-phase chemistry was simulated with the Carbon Bond

2005 (TUCL05) chemical mechanism (Yarwood et al., 2005;Whitten et al., 2010;Sarwar et al., 2012). Aerosol chemistry and partitioning was simulated using the aerosols 6 (AERO6) module with different models to represent OA (described below). United States anthropogenic emissions were based on the EPA's 2011v1 modeling platform (http://www.epa.gov/ttnchie1/net/2011inventory.html) and biogenic emissions were estimated using the Biogenic Emission Inventory (BEIS) version 3.14 model (Carlton and Baker, 2011).

Gridded meteorological inputs for CMAQ and SMOKE were generated using version 3.1 of the WRF model (Skamarock et al., 2008). The first 11 days of the simulation were excluded from the analysis to minimize the influence of initial conditions. Boundary conditions were provided by a 36-km continental U.S. CMAQ simulation from the same time period.

### 2.2 OA Model

The OA model used here builds on the volatility basis set (VBS) implementation in CMAQ (Koo et al., 2014) and is referred to as the VBS-IVOC model. The novel aspects of this work are the implementation of updated organic emissions profiles that explicitly account for IVOC emissions from gasoline and diesel sources and experimentally constrained parameterizations of Jathar et al. (2014) for the SOA production from IVOC emissions.

In the VBS-IVOC model, we extended the work of Baker et al. (2015) and Woody et al. (2016), both of which evaluated different OA models in CMAQ using the CalNex data. Baker et al. (2015) evaluated the standard OA module in CMAQ (Carlton et al., 2010). Woody et al. (2016) evaluated the VBS version of CMAQ as implemented by Koo et al. (2014), which treated POA emissions as semi-volatile and reactive and

accounted for SOA production from VOCs and IVOCs and multigenerational oxidation of aged products. The VBS-IVOC model was the same as the VBS model of Woody et al. (2016) except for the treatment of gasoline and diesel sources. To facilitate direct comparison between the different models, all three studies

(this work, Baker et al. (2015) and Woody et al. (2016)) used the same CTM (CMAQ v5.02), emissions inventory (except for the modifications described below) and meteorology inputs. However, Baker et al. (2015) used a different gas-phase chemical mechanism (SAPRC07b). We refer to the Baker et al. (2015) treatment of OA and the model results as the 'Traditional' model and we refer to the Woody et al. (2016) treatment of OA and the model results as the 'VBS' model.

The VBS version of CMAQ includes four distinct volatility basis sets to separately track different classes of OA: anthropogenic POA, anthropogenic SOA, biogenic SOA, and biomass burning POA (Koo et al., 2014). The VBS-IVOC model extended CMAQ with three additional basis sets for POA from gasoline sources, diesel sources, and cooking activities to provide POA source apportionment (Woody et al., 2016). Each basis set has five volatility bins with different effective saturation concentrations (C*): non-volatile and logarithmically distributed bins from $10^0$ to $10^3$ µg m$^{-3}$ at 298 K. The gas-particle partitioning of semi-volatile organic compounds in each basis set is assumed to be in equilibrium and to form a quasi-ideal solution with all of the OA.

**Emissions:** In the VBS-IVOC model, we used emission inventories developed by Baker et al. (2015) and modified by Woody et al. (2016) for use with the VBS model. In this section we briefly describe the VBS inventory of Woody et al. (2016), focusing on the updates to gasoline and diesel organic emissions used in the VBS-IVOC model.

We (in the VBS-IVOC model) and Woody et al. (2016) used the same semi-volatile POA emissions. These were estimated by redistributing the non-volatile POA emissions of Baker et al. (2015) into the VBS. For gasoline and diesel exhaust and biomass burning, this redistribution was done using the source-specific volatility distributions of May and coworkers (May et al., 2013b, c;May et al., 2013a). Cooking emissions were redistributed using an approximation developed by Woody et al. (2016) based on thermodenuder measurements made with cooking emissions and ambient measurements made during MILAGRO (Huffman et al., 2009). For all other sources, the volatility distribution of Robinson et al.(Robinson et al., 2007) was used to map the existing POA emissions into the VBS.

In the VBS-IVOC model, we used new VOC speciation profiles for tailpipe emissions from gasoline and diesel sources (Table S.1). These speciation profiles were applied to the emissions inventory of Baker et al. (2015). Therefore, the VBS-IVOC model had the same mobile source emission rates as Baker et al. (2015) but with different organic speciation. For all gasoline sources (on and off-road), the VOC speciation was

based on fleet-averaged data from May et al. (2014), which reported emissions of 202 unique species measured during chassis dynamometer testing of 68 light-duty gasoline vehicles operated over the Cold Unified Cycle (UC) using gasoline that met California summertime specifications (five of the vehicles were also run on the Freeway, Arterial and Hot UC cycles). For on- and off-road diesel vehicles, the VOC speciation was derived from the EPA SPECIATE profile for on-road heavy-duty diesel vehicles (Profile number 8774); the same diesel emissions profile was used in Baker et al. (2015) and Woody et al. (2016). All VOCs were mapped to CB05 model species using EPA's speciation tool, which lumps unique organic compounds to a representative model species that are similar in terms of their reactivity and reaction chemistry (Eyth et al., 2006;Carter, 2008).

For gasoline and diesel sources, we estimated the IVOC emissions in the VBS-IVOC model based on the gas-phase carbon-balance analysis of Jathar et al. (2014), who found that unspeciated organic compounds (assumed to be mainly IVOCs) contributed, on average, 25% and 20% of the non-methane organic gas (NMOG) emissions from gasoline and diesel vehicles respectively. IVOCs were included in the VBS-IVOC model by reapportioning the existing non-methane organic gas (NMOG) emissions between VOCs and IVOCs (effectively renormalizing the VOCs described above). Therefore, unlike previous VBS models such as Woody et al. (2016) where IVOC emissions were added to the NMOG emissions, no new NMOG emissions were added to the VBS-IVOC model for the gasoline and diesel sources. In addition, gasoline and diesel POA emissions in the C* bins of $10^3$ and $10^4$ $\mu g\ m^{-3}$ (organic compounds that exist in the vapor phase in the atmosphere; 32% of gasoline and 35% of diesel POA emissions) were reclassified as IVOCs, consistent with the parameterization of Jathar et al. (2014).

Following Robinson et al. (2007), IVOC emissions for all other sources (non-gasoline and diesel) were assumed to be 1.5 times the POA emissions (Woody et al. (2016) assumed this for all sources). Some of the IVOCs, as defined here, may have already been included in the original emissions profile as ALK5 and UNK, however, Pye and Pouliot (2012) show that these emissions are very likely underestimated and, therefore, do not pose a serious problem of double counting SOA precursors.

To illustrate the effects of these changes, Figure 1 plots the POA and SOA precursor emissions (BTEX (all aromatics), ALK5 (long alkanes) and IVOCs) from all gasoline and diesel sources in Los Angeles and Orange counties aggregated over the entire simulation period (May 4 to June 30, 2010). Table S.2 in the supplementary material lists the emissions for on- and off-road gasoline and diesel use, all other sources, and biogenic sources. Here, gasoline and diesel sources include both on- and off-road applications.

The magnitude of the POA emissions was identical between all three models with the exception that some of the POA emissions were reclassified as IVOCs in the VBS-IVOC model as described earlier. The BTEX emissions were identical between the Traditional and VBS models, but lower in the VBS-IVOC model because we renormalized the NMOG emissions to account for IVOCs. The Traditional model did not include IVOC emissions. The IVOC emissions in the VBS-IVOC model were a factor of four higher for gasoline sources than in the VBS model of Woody et al. (2016), but 20% lower for diesel sources. Taken together, the BTEX, ALK5 and IVOC emissions (sum of all anthropogenic SOA precursors) were somewhat higher (40%) in the VBS-IVOC model compared to the VBS model for gasoline sources and slightly lower (5%) for diesel sources. Therefore, discounting for differences in reaction rate constants and SOA mass yields, we expected roughly similar SOA production from gasoline and diesel sources between the VBS and VBS-IVOC simulations. In all models, gasoline sources have substantially larger organic emissions than diesel sources (e.g. 3.7, 42, 35 and 16 times more POA, BTEX, ALK5 and IVOC for the VBS-IVOC model, respectively); therefore we anticipated much higher SOA production from gasoline sources than from diesel sources.

**SOA formation:** SOA production from VOCs was simulated using the parameterizations of Murphy and Pandis (2009) except for toluene (Hildebrandt et al., 2009). SOA production from aromatics (toluene, xylene, and benzene), isoprene, and monoterpenenes have high- and low-$NO_x$ yields; there was no $NO_x$ dependence in the SOA yield from sesquiterpenes and IVOCs. Emissions profiles for VOCs, IVOCs and their SOA yields, specific to gasoline and diesel tailpipe emissions, are presented in Table S.1.

IVOC emissions from gasoline and diesel sources were represented separately using two (one for gasoline and one for diesel) gas-phase species in the chemical mechanism (CB05-TUCL) and the parameterizations of Jathar et al. (2014) were used to estimate the SOA production from the IVOC oxidation. Briefly, the IVOCs reacted with the hydroxyl radical (OH) to form a set of semi-volatile products distributed in the VBS (Table S.1). The stoichiometric mass yields for each product were determined by fitting the SOA production measured in smog chamber experiments performed with diluted vehicle exhaust (Jathar et al., 2014). Following Woody et al. (2016), for all other sources (i.e. not gasoline and diesel) SOA production from IVOCs was based on the published yields for the SAPRC ARO2 model species from Murphy and Pandis (2009).

SOA formed from VOCs and IVOCs was aged via reactions of the organic vapors with OH using a rate constant of $2 \times 10^{-11}$ $cm^3$ molecules$^{-1}$ s$^{-1}$. These aging reactions formed products with a vapor pressure reduced

by one order of magnitude. Biogenic SOA was not aged based on previous findings that aging reactions over-predicted OA concentrations in rural areas (Fountoukis et al., 2011;Lane et al., 2008;Murphy and Pandis, 2009). Semi-volatile POA vapors from all sources were aged using the scheme of Robinson et al. (2007) – gas-phase reactions with the OH using a rate constant of $4 \times 10^{-11}$ cm$^3$ molecule$^{-1}$ s$^{-1}$, which lowered volatility by an order of magnitude (Robinson et al., 2007). Finally, the aging reactions also shifted a portion (~10%) of the POA vapors to the anthropogenic SOA basis set to maintain O:C ratios (Koo et al., 2014). OH was artificially recycled in the IVOC oxidation and all aging reactions to prevent double counting and impacts to the gas-phase chemistry of the underlying chemical mechanism (Koo et al., 2014). The rate constants used for the aging reactions were not well constrained and are likely to vary with C* and O:C.

## 3 Results

Although the simulation domain covers the entire state of California, we focused our analysis on model predictions over southern California and the metropolitan area of Los Angeles. This region is the second most populated area in the US, has historically had severe air pollution problems, and was the focus of a major air quality campaign (CalNex) during the simulation period.

### 3.1 Spatial distribution of OA

Figure 2 shows maps of average predicted concentrations of total OA (POA+SOA) from the VBS-IVOC model for the following sources: (a) all, (d) gasoline, (e) diesel, (f) biogenic and (g) other. In addition, Figure 2 also plots the predicted ratios of (b) POA to OA and (c) SOA to OA. Average predicted concentrations of OA in southern California ranged between 1.5 and 3 µg m$^{-3}$ with POA accounting for slightly more than half of the OA in source regions such as downtown Los Angeles (a 'source' region was defined as one with high anthropogenic emissions of species such as POA) and SOA dominating in non-source regions and off the coast.

Gasoline sources were predicted to contribute ~35% of the inland OA while diesel sources contributed less than 3% (for details see Section 4). The predicted gasoline OA exhibited a slightly different spatial pattern than total OA, with higher downwind concentrations near Riverside than those near central Los Angeles, reflecting the importance of atmospheric production of SOA. As expected, biogenic SOA was more important outside of the urban areas contributing 5% of total OA in urban areas versus 10-20% in non-urban areas. Other OA contributed slightly more than half of all OA in the urban areas. Other OA was dominated by cooking POA, biomass burning POA and other anthropogenic SOA (see Figure 4 for contributions of these sources in Pasadena).

## 3.2 Model Evaluation Using OA Mass and Composition Measurements

The VBS-IVOC model was evaluated using measurements made at the Chemical Speciation Network (CSN) and the CalNex Pasadena ground sites. Figure 3(a) compares predicted daily-averaged OA mass concentration to measurements of organic carbon (OC) made at six CSN sites in California (Fresno, Bakersfield, Central Los Angeles, Riverside, El Cajon and Simi Valley). Figure 3(b) compares predicted daily-averaged OA concentrations to measurements made using a high-resolution aerosol mass spectrometer (HR-AMS) in Pasadena (Hayes et al., 2013). The CSN measurements were multiplied by an OA:OC ratio to account for the non-carbon species associated with organic carbon (Turpin and Lim, 2001). While ambient OA:OC ratios can range between 1.4 and 2.3 (Aiken et al., 2008), we used a value of 1.6 in this work based on previous estimates used for filter-based measurements (e.g., (Cappa et al., 2016)). This value was consistent with the OA:OC ratio of 1.7±0.5 estimated by Hayes et al. (2013) in Pasadena.

Predictions from the VBS-IVOC model were slightly lower than the filter-based measurements at the CSN sites, similar to other studies (Simon et al., 2012). The fractional bias and fractional error versus CSN sites was -23% and 43%, respectively. At the CSN sites, predictions from the VBS-IVOC model were marginally better at the southern California sites (Central LA, Riverside, El Cajon, Simi Valley, Pasadena) than the central California sites (Fresno, Bakersfield). This may be due to sources related to oil and gas production and agricultural activity being more important in central California (Gentner et al., 2014).

Figure 3(b) indicates predictions from the VBS-IVOC model were a factor of three lower than the HR-AMS OA data at the Pasadena site. It is unclear why the model performs much better at numerous CSN sites than the Pasadena site. One possibility is that the Pasadena site is influenced by local sources and transport that is not captured by the model at a 4 km resolution.

OA mass concentrations are only one measure for evaluating model performance. Given the myriad sources of and complexity in SOA production, a model can predict the right absolute OA concentration for the wrong reason. Therefore, it was important to evaluate the model against OA composition. Figure 4 compares predicted POA and SOA mass fractions to results from a positive matrix factorization (PMF) analysis of HR-AMS measurements made in Pasadena (Hayes et al., 2013). Since the absolute OA concentrations as measured with the HR-AMS were under-predicted (Figure 3(b)), we focused on OA mass fractions. Mass fractions only allow for a qualitative comparison of the OA composition and any differences in the modeled and measured mass fractions cannot be interpreted as an under- or over-prediction in the absolute mass concentration.

Figure 4 compares model predictions to hydrocarbon-like OA (HOA), cooking OA (COA) and oxygenated OA (OOA) factors derived from the ambient HR-AMS data (Hayes et al., 2013). The AMS HOA factor is typically associated with POA from motor vehicles and other fossil fuel sources. Therefore, in this work, it

is compared against predictions of POA from gasoline and diesel sources. The AMS COA factor is associated with primary cooking emissions and is compared against predictions of POA from cooking sources. The AMS OOA factor is associated with SOA and is compared against predictions of total SOA; the model did not resolve SOA by degree of oxygenation and hence we have not compared predictions to the individual HR-AMS-derived semi-volatile OOA (SV-OOA) and low-volatility OOA (LV-OOA) factors.

Before discussing the normalized composition predicted by the VBS-IVOC model, we briefly describe the findings from Woody et al. (2016) who carefully compared the predictions of absolute concentrations of the VBS model to the PMF factors estimated from the ambient HR-AMS measurements. Woody et al. (2016) found that (i) the predicted cooking-related OA concentrations compared well with the COA factor during

the morning but were low in the afternoon and late night, (ii) non-cooking POA concentrations compared well with the HOA factor except during the afternoon when it was underpredicted, and (iii) predicted SOA concentrations matched the diurnal profile of the OOA factor but were a factor of 5 lower during all times of the day.

Figure 4 shows that the VBS-IVOC model better predicts the POA-SOA split than the Traditional model. For the VBS-IVOC model, the POA-SOA split was 1:1 versus ~20:1 for the traditional model. The measurement-based factor analysis estimated a POA-SOA split of 1:2. For the Traditional model, SOA contributed less than 3% of the total OA.

In Figure 4, we show that the predicted gasoline+diesel POA fraction compared well with the HR-AMS HOA fraction while the predicted cooking POA fraction was over-predicted compared to the HR-AMS COA fraction; Woody et al. (2016) hypothesized that the VBS model (which has the same treatment for cooking OA as the VBS-IVOC) likely under-predicted cooking POA emissions based on a comparison of absolute cooking OA concentrations. For the VBS-IVOC model, about 6% of the OA was from biomass burning while

Hayes et al. (2013) were unable to determine a biomass burning factor in their PMF analysis of ambient data. The SOA fraction predicted by the VBS-IVOC model was about 35% lower than the estimated OOA fraction. It is unclear if the predicted non-mobile, non-cooking and non-biomass burning POA (which in Pasadena accounts of ~9% of the OA) should be combined with SOA before being compared with ambient OOA factor.

The non-mobile, non-cooking and non-biomass burning POA (or anthropogenic (other) POA) category here includes sources such as stationary fuel combustion (e.g., natural gas combustion), surface coatings (e.g., metal coating), mineral processes (e.g., concrete production), road dust and managed burning (e.g., prescribed burns). Unfortunately, the composition of the POA emitted from these sources is not well understood and needs to be investigated by future work.

Although predictions from the VBS-IVOC model were much better than the Traditional model for the POA-SOA split and the fractional source contribution/composition of OA, in Figure 3(b) we show that predictions from the VBS-IVOC model were substantially lower than the absolute concentrations measured by the HR-AMS. Future research should explore higher resolution simulations (<1 km) for the Los Angeles area, in addition to improving estimates of POA emissions (e.g., cooking) and improved representations for SOA formation (e.g., higher SOA yields when accounting for vapor wall-losses in chambers).

**3.3 Model Evaluation Using IVOC Measurements**

A novel aspect of the VBS models (VBS and VBS-IVOC) is that they track IVOCs, an important class of SOA precursors (Jathar et al., 2014). Campaign-averaged predictions of IVOC concentrations are compared in Figure 5 against IVOC measurements at the Pasadena ground site made by Zhao et al. (2014). This was the first time 3-D model predictions of IVOCs have been compared against ambient measurements. The VBS-IVOC model did not simulate secondary production of IVOC species (for lack of data) and hence model predictions in Figure 5 only include primary emissions of IVOCs. The IVOC measurements shown in Figure 5 are split into two categories: primary and oxygenated. Zhao et al. (2014) attributed the measured primary IVOCs to emissions from mobile sources (gasoline+diesel) and oxygenated IVOCs to primary sources and those formed in the atmosphere.

Predicted gasoline and diesel IVOC concentrations (3.9 $\mu g\ m^{-3}$) from the VBS-IVOC model were 35% lower when compared to the hydrocarbon IVOCs concentrations measured by Zhao et al. (2014) (6 $\mu g\ m^{-3}$). In contrast predictions from the VBS model were a factor of 4 lower than the measurements, which highlights the improved representation of IVOCs in the VBS-IVOC model; the traditional model predicted essentially no IVOCs. The under-prediction of VBS-IVOC could partly be a result of the inability of the model with a 4 km horizontal resolution to capture the location-specific concentrations at Pasadena. The model-measurement comparison suggests that the VBS-IVOC model reasonably simulated the emissions, transport and chemistry of IVOCs from mobile sources. Furthermore, the VBS-IVOC model predicted that the majority of the hydrocarbon IVOCs originated from gasoline sources. Coincidentally, the predicted IVOC sum for other anthropogenic sources and biomass burning (4.3 $\mu g\ m^{-3}$) compared well with the measured oxygenated

IVOCs (4.1 µg m$^{-3}$). Given the uncertainty in the model emissions of IVOCs for non-mobile sources (POAx1.5), the comparison with oxygenated IVOCs needs to be explored in future work.

**3.4 Model Inter-comparison for OA**

We compared predictions from the VBS-IVOC model to OA predictions from Baker et al. (2015) and Woody et al. (2016) who simulated air quality in California during CalNex. Figure S.2 presents maps of averaged concentrations and ratios of POA, SOA and total OA (POA+SOA) from the Traditional and VBS-IVOC models. The results were qualitatively similar to earlier VBS implementations (Fountoukis et al., 2014;Hodzic et al., 2010;Ahmadov et al., 2012;Shrivastava et al., 2011;Tsimpidi et al., 2009) and previous comparisons between VBS and Traditional-like models (Robinson et al., 2007;Shrivastava et al., 2008;Woody et al., 2016;Jathar et al., 2011). In the VBS-IVOC simulation, total OA concentrations were lower in source regions (~50%) but ~20-40% higher away from sources than the Traditional model. The decrease in source regions was due to POA evaporation while an increase away from sources resulted from enhanced SOA production. The OA predicted by the Traditional model was dominated by POA (1-3 µg m$^{-3}$) with very little SOA (0.2-0.4 µg m$^{-3}$) while the OA predicted by the VBS-IVOC model had equal proportions of POA and SOA.

Figure S.3 compares predictions of the VBS and VBS-IVOC models, including average concentrations and ratios of POA, SOA and total OA (POA+SOA). The results were surprisingly similar. POA concentrations in the VBS-IVOC model were slightly lower (~10%) in source regions and lower still in non-source regions (~20%) than the VBS model. The SOA concentrations were nearly identical and both models predicted more spatially uniform OA concentrations compared to the Traditional model. The modest differences in POA and SOA likely resulted from a combination of the following three reasons: (1) the magnitude of the total SOA precursor emissions in the VBS and VBS-IVOC models were basically the same (see BTEX, ALK5 and IVOC emissions data in Table S.2 for all sources), (2) gasoline and diesel sources contributed only 30%-40% of the predicted OA concentrations in southern California (see Section 5 for a detailed discussion) and (3) a majority of the SOA predicted in southern California arises from aging reactions.

Although the VBS and VBS-IVOC models contain very different representations of mobile source emissions, these emissions contributed, on average, to slightly more than one-third of the total OA in southern California (see Section 4). Therefore, the updates used in the VBS-IVOC model had a limited influence in affecting the overall OA burden. Strict regulations have dramatically reduced emissions from motor vehicles over the past three decades, which has both improved air quality and increased the relative importance of other sources to

OA (McDonald et al., 2015). For example, compared to mobile sources, cooking remains a possibly important, yet understudied, source of fine particle pollution in urban airsheds.

The similarity between predictions from the VBS and VBS-IVOC models was also due to the importance of aging reactions. Both models used the same aging scheme applied to POA and SOA vapors (for more details, see Koo et al. (2014)) To quantify its contribution to predicted SOA concentrations, we ran the VBS-IVOC model with aging reactions turned off; these results are plotted in Figure S.4. Without aging, total predicted OA was nearly halved and SOA concentrations were significantly reduced (more than a factor of five in source regions, factor of 10 to 20 in terrestrial non-source regions and up to a factor of 40 over the ocean). Given that mobile sources contributed only about one-third of the total OA and that aging reactions significantly enhanced OA concentrations, it appears that modest differences in the emissions and yield potential of SOA precursors between the VBS and VBS-IVOC models had a limited effect on the OA burden.

**4 Gasoline versus Diesel Source Contributions to OA**

Recent analyses of the CalNex data have led to conflicting conclusions about the contribution of gasoline and diesel sources to OA in southern California (Bahreini et al., 2012;Gentner et al., 2012;Ensberg et al., 2014;Hayes et al., 2013;Zotter et al., 2014;Hayes et al., 2015). The source-resolution implemented in the VBS-IVOC model allowed for an assessment of the absolute and relative importance of gasoline and diesel sources to OA in southern California. In Figure 6, we plot the campaign-averaged OA concentrations attributable to gasoline and diesel use. The SOA production from VOCs emitted by gasoline and diesel sources was not tracked separately in the model. Here, the SOA from VOCs was estimated based on the contribution of gasoline and diesel sources to the emissions of VOC precursors (BTEX and ALK5) in Los Angeles and Orange counties.

In Pasadena, predictions from both VBS models showed that gasoline sources contributed ~7 to 8 times more OA than diesel sources (Figure 6(a)), which was somewhat lower than other inland locations in southern California (Figure 6(b)). Domain-wide, the median predicted gasoline contribution to OA was 13 times that of diesel. At Pasadena, predictions from the VBS-IVOC model showed that gasoline contributed 20 times more SOA than diesel. Both VBS models predicted that the combined (gasoline and diesel) POA-to-SOA split was ~1:3 implying that the contribution of gasoline and diesel sources to ambient OA strongly depends on SOA production and not directly-emitted POA. Based on results from the VBS-IVOC model, gasoline sources produced more SOA than POA (SOA~3.6xPOA) while diesel sources produced less SOA than POA (SOA~0.5xPOA). Comparison of the POA predictions from the VBS-IVOC model to ambient measurements

made by Ban-Weiss et al. (2008) suggests that the on-road gasoline POA in the model may be over-predicted by a factor of 2, although this over-prediction does not significantly change the gasoline/diesel contribution to OA.

Our predictions for the large contribution of gasoline vehicle exhaust to SOA were consistent with the weekday/weekend analysis of Bahreini et al. (2012) and qualitatively similar to the findings of Zotter et al. (2014) and Hayes et al. (2013). However, Hayes et al. (2015) predicted a much larger contribution of diesel sources to SOA than this work (only1.5 to 2 times lower than gasoline), which can mostly be attributed to the differences in emissions inputs for S/IVOC emissions. (Hayes et al. (2015) estimated that 44-92% of the
SOA comes from S/IVOCs). Hayes et al. (2015) estimated S/IVOC emissions by scaling POA emissions based on Schauer et al. (1999) and using the volatility distribution from Robinson et al. (2007). The POA scaling data are from two medium duty vehicles manufactured more than two decades ago and the volatility data are from a single diesel engine manufactured a decade ago. In contrast, our work used a much more comprehensive dataset to determine S/IVOC emissions from gasoline and diesel sources. Finally, the
emissions inventory (see Table S.1) suggests that the Traditional model (with a non-volatile POA and little SOA production) would have predicted that gasoline sources contribute four times more OA than diesel sources.

We also investigated the sensitivity of the VBS-IVOC predictions to uncertainty in diesel IVOC emissions.
Zhao et al. (2015) recently directly measured the IVOCs from emissions of on-road diesel engines. They found that IVOCs could contribute up to 60% of the NMOG emissions, which was much greater than the 20% used here. To explore the implications of the findings of Zhao et al. (2015), we performed two additional sensitivity simulations with the VBS-IVOC model where we scaled IVOC emissions from diesel sources by a factor of 3 and 5, which were effectively equivalent to IVOC-to-NMOG ratios of 0.6 and 1.0 respectively.
For these simulations, additional IVOC mass was added to the inventory.

Results from the IVOC sensitivity simulations are also shown in Figure 6(a). We found that increasing the IVOC emissions proportionally increased the OA contribution from diesel sources. However, even if all of the NMOG emissions from diesel were IVOCs (an upper bound estimate), gasoline-related OA still
dominated OA from diesel sources. A factor of 5 increase in IVOC emissions only resulted in a 0.025 µg m$^{-3}$ increase in total OA mass concentration. Therefore, uncertainty in the diesel IVOC emissions did not appear to alter the model-measurement comparison discussed earlier.

Figure 6(c) shows the cumulative distribution for the fractional contribution of gasoline and diesel sources to total OA across southern California. Gasoline sources contributed much more to the total OA (median contribution of 35%) than diesel sources (median contribution of 2.6%) over southern California (Figure 6(c)). Together, mobile sources (gasoline and diesel use) contributed ~30-40% (10th-90th percentile) of the predicted OA concentration in southern California. Therefore, mobile sources remain the single most important source despite decades of increasingly strict emissions controls. The balance of the OA was from cooking POA (median contribution of 10%), biogenic SOA (median contribution of 10%) and all other anthropogenic sources (median contribution of 40%, which includes SOA from cooking sources). Gasoline sources were still predicted to be the largest single source category. This finding partially supports the conclusion of Ensberg et al. (2014) that mobile sources do not contribute to the majority of OA in southern California and potentially explains why the updates only modestly changed the overall model predictions.

Figure 6(a) resolves the OA contributions based on the precursor class at the Pasadena site. The VBS-IVOC model predicted that IVOCs, particularly from gasoline vehicles, formed almost as much SOA as VOCs (long alkanes and single-ring aromatics). This was in contrast to Jathar et al. (2014), who found that unspeciated precursors (or IVOCs) were approximately a factor of 4 larger than VOCs in forming SOA in chamber experiments. One possible explanation for this difference was that Jathar et al. (2014) did not account for the effects of continued aging of IVOC oxidation products on OA concentrations. Simulations with the VBS-IVOC model with aging reactions turned off (discussed in Section 3.4) indicate that aging enhanced VOC SOA by a factor of 14 but enhanced IVOC SOA only by a factor of 3-5. The different enhancements were caused by different product distributions for VOC and IVOC SOA in volatility space. This underscores the uncertainty in the treatment of aging reactions.

Platt et al. (2014) and Gordon et al. (2013) have recently argued that off-road sources, especially those powered using two-stroke engines, can be a large contributor to fine particle pollution in cities. In the inventory of Baker et al. (2015), which was used in this work, off-road sources contributed to ~40% of the NMOG and ~40% of the POA emissions from mobile sources. Given their substantial emissions, it is critical then that emissions rates from these sources be accurately represented in large-scale models. Only one study so far has reported VOC and IVOC emissions profiles from off-road engines. May et al. (2014) have found that two-stroke off-road gasoline engines have similar emissions profiles as on-road gasoline engines, but that the four-stroke off-road gasoline engines had much higher IVOC fractions than on-road gasoline engines. However, Platt et al. (2014) have shown that most of the SOA produced from two-stroke off-road gasoline engines can be explained by the emissions and oxidation of aromatic compounds and they did not find IVOCs

to be an important precursor of SOA. In our work, we have assumed that the VOC speciation, IVOC fraction of NMOG, and the SOA parameterization for IVOCs were identical between the on- and off-road mobile sources. Given the uncertainties, these assumptions may need to be examined in detail in future work.

**5 Conclusions**

In this work, we developed an updated version of the CMAQ model that included revised estimates of (i) VOC and IVOC SOA precursors from gasoline and diesel sources and (ii) experimentally constrained parameterizations for SOA production from IVOCs. Predictions of OA mass concentrations from the updated model (VBS-IVOC) slightly under-predicted daily-averaged, filter-based measurements at CSN sites in California during May and June 2010 (fractional bias=-23% and fractional error=43%) but were a factor of

three lower than aerosol mass spectrometer-based measurements made at Pasadena as part of the CalNex campaign. The Pasadena site may have been influenced by local sources and transport not captured by the model at a 4 km resolution. We recommend future modeling studies to be performed at higher resolution.

When compared to a Traditional model of OA in CMAQ that includes a non-volatile treatment of POA and

no SOA from IVOCs, the VBS-IVOC model produced different spatial patterns of OA with lower (~50%) concentrations in source regions but higher (~20-40%) concentrations away from the sources. The VBS-IVOC model in comparison to the Traditional model improved predictions of the sources and composition of OA. These findings are consistent with previous comparisons between Traditional- and VBS- models and highlight the importance of the use of an OA model that includes semi-volatile and reactive POA and SOA

formation from IVOCs.

Predictions of OA from the VBS-IVOC model were similar to those from a recently released research version of CMAQ (VBS) that included semi-volatile POA and SOA formation from IVOCs (Woody et al., 2016). The predictions of these two models were similar for three reasons. First, the VOC and IVOC updates in this

work, surprisingly, did not substantially alter the total emissions of SOA precursors in southern California (although the VOC-IVOC composition was different between the two models for gasoline sources). Second, mobile sources only accounted for slightly more than one-third of the total OA in southern California and hence updates to the emissions and SOA production from mobile sources had a limited influence on the total OA burden. And third, and most important, was that both models predicted that multigenerational aging of

vapors in equilibrium with OA was a major source of SOA. Both models used similar aging mechanisms that were conceptually based on the work of Robinson et al. (2007), which assumed a constant reaction rate constant and only allowed for the formation of functionalized, lower-volatility products. However, reaction

rates may vary with C* and O:C of the OA and fragmentation reactions can be increasingly important at longer time scales (Kroll et al., 2011). Furthermore, existing aging mechanisms have not been constrained with laboratory data. This implies that the OA predictions, despite the substantial new data, are poorly constrained as one moves downwind of source regions. Murphy and Pandis (2009) have reported improved

model performance when aging reactions were turned off for biogenic SOA. Recently, Jathar et al. (2016) proposed that laboratory chamber experiments that were used to parameterize SOA production may already include products from some aging reactions, raising concerns about double counting. Although some work has been done to understand the aging of biogenic SOA (Donahue et al., 2012;Henry and Donahue, 2012), future laboratory work needs to be directed in understanding the role of aging of OA vapors formed from

anthropogenic sources on the mass and properties of OA.

For the first time, we compared model predictions to ambient measurements of IVOCs. The new VBS-IVOC model better predicted the ambient IVOC concentrations compared to the Traditional and VBS models, This suggests that the updated model reasonably simulated the emissions, transport and chemistry of IVOCs from

mobile sources. However, the model representation of IVOCs from non-mobile sources remains poorly constrained and needs to be explored through future emissions, laboratory and modeling studies.

Finally, the VBS-IVOC model predicted that mobile sources accounted for 30-40% of the OA in southern California, with half of the OA being SOA. The diurnal variation of OA in Pasadena supports the hypothesis

that substantial OA is produced through photochemical reactions vs. primary emissions (Hayes et al., 2013). Gasoline-powered sources contributed 13 times more OA than diesel-powered sources and sensitivity simulations indicated that these findings were robust to changes in diesel emissions. Model predictions suggested that half of the mobile source SOA was formed from the oxidation of IVOCs, which demonstrates the importance of including IVOCs as an SOA precursor. However, the relative contribution of VOCs and

IVOCs to SOA formation was sensitive to the inclusion of aging reactions. While both laboratory and field evidence indicate that aging is an important atmospheric process, it is unclear if and by how much aging enhances OA over regional scales and whether aging chemistry varies by precursor and source (Jathar et al., 2016). For these reasons, the relative importance of VOC and IVOC SOA precursors and the source apportionment presented here is a first estimate and will likely evolve as we develop better models to simulate

the dependence of aging on SOA formation.

## 6 Acknowledgments

We would like to acknowledge the contributions of Matti Maricq and Timothy Wallington of the Ford Motor Company, Rory MacArthur of the Chevron Corporation, Hector Maldonado of the California Air Resources Board and the Coordinating Research Council Real World Vehicle Emissions and Emissions Modeling
Group and Atmospheric Impacts Committee. This research was supported by the US Environmental Protection Agency National Center for Environmental Research through the STAR program (Project RD834554) and the Coordinating Research Council (Project A-74/E-96). The views, opinions, and/or findings contained in this paper are those of the authors and should not be construed as an official position of the funding agencies. Disclaimer: Although this work was reviewed by EPA and approved for publication,
it may not necessarily reflect official agency policy.

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

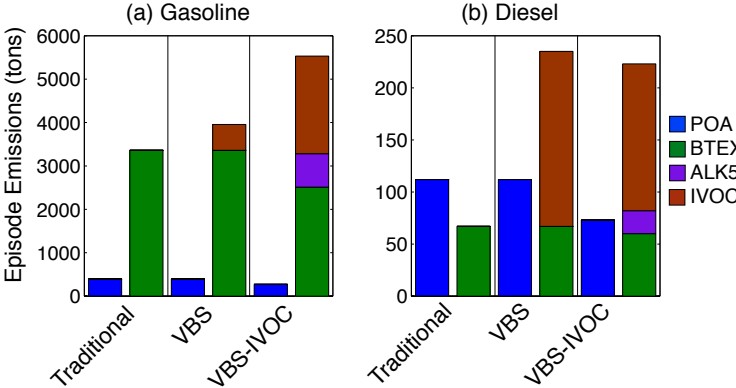

**Figure 1: Total emissions from May 4 to June 30, 2010 for POA, BTEX (aromatics), ALK5 (long alkanes) and IVOCs for gasoline and diesel sources in the Los Angeles and Orange Counties for the three OA models: Traditional, VBS and VBS-IVOC.**

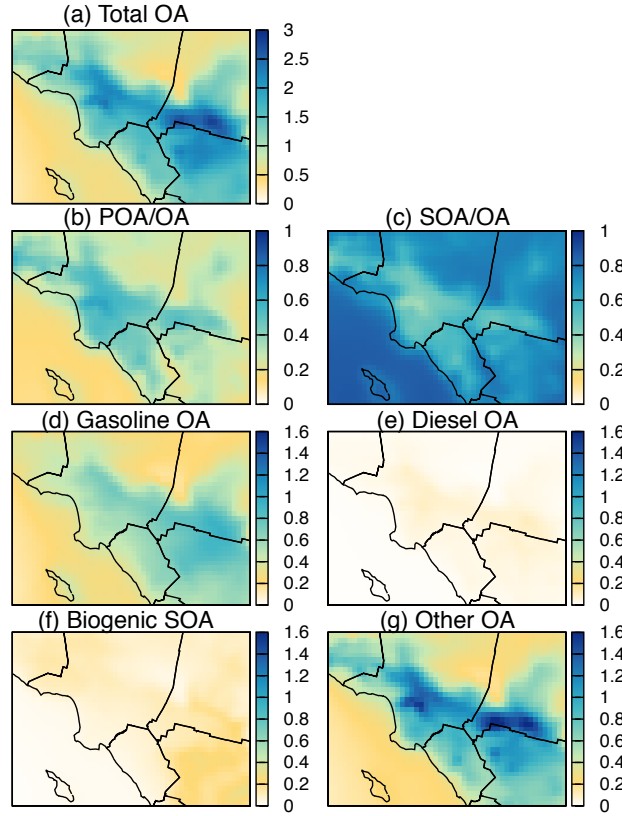

**Figure 2: Averaged predictions from the VBS-IVOC model for (a) total OA ($\mu g\ m^{-3}$), (b) POA fraction, (c) SOA fraction, (d) total gasoline OA ($\mu g\ m^{-3}$), (e) total diesel OA ($\mu g\ m^{-3}$), (f) biogenic SOA ($\mu g\ m^{-3}$) and (g) other OA ($\mu g\ m^{-3}$) over southern California.**

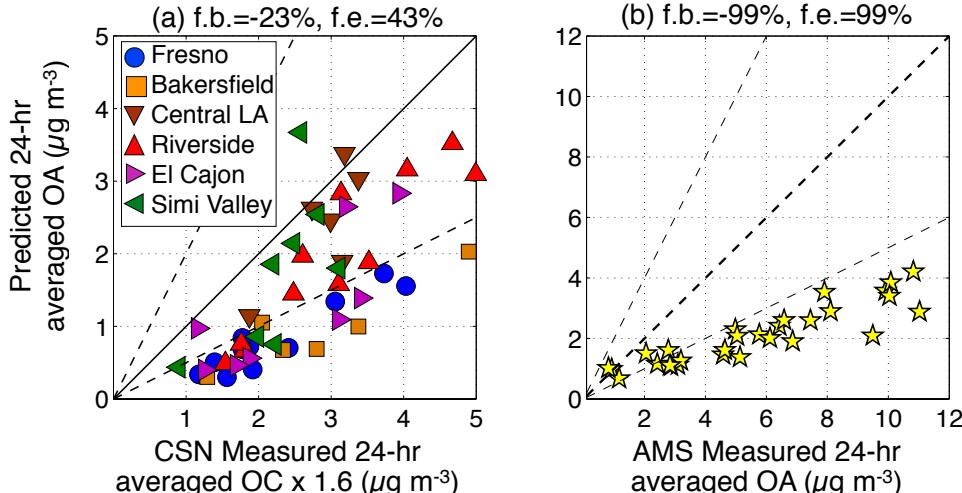

**Figure 3: Scatter plot of VBS-IVOC OA predictions versus 24-hr measurements from (a) filters collected at sites in the Chemical Speciation Network (CSN) and (b) HR-AMS measurements at the Pasadena ground site during the CalNex campaign. In panel (a) the model-measurement comparison is for six sites in California (Fresno, Bakersfield, Central Los Angeles, Riverside, El Cajon and Simi Valley). f.b. is the fractional bias ($\frac{1}{N}\sum_{i=1}^{N}\frac{P-M}{\frac{P+M}{2}}$) and f.e. is the fractional error ($\frac{1}{N}\sum_{i=1}^{N}\frac{|P-M|}{\frac{P+M}{2}}$); $P$ is the predicted value, $M$ is the measured value and $N$ is the sample size.**

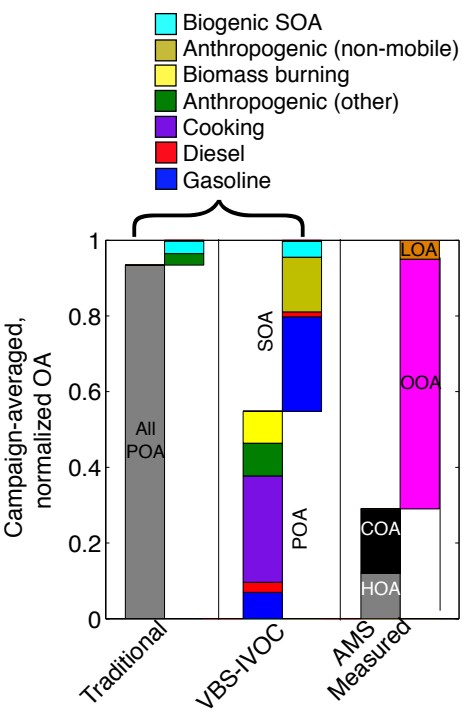

**Figure 4: Averaged, normalized composition of OA at the Pasadena ground site as predicted by the Traditional and VBS-IVOC models. Predictions are compared to PMF factors derived from ambient HR-AMS data collected in Pasadena Hayes et al. (2013).**

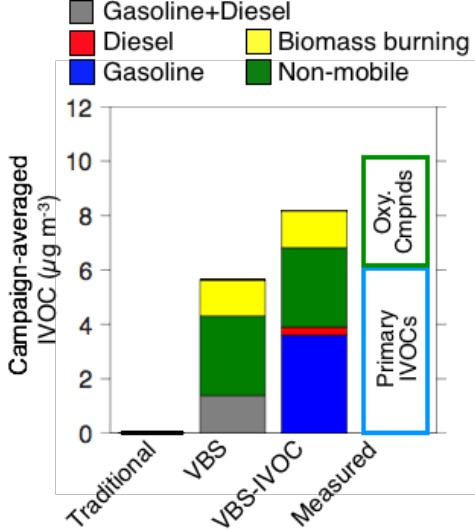

**Figure 5: Comparison of predicted and measured campaign-averaged IVOC concentrations at the Pasadena ground site. Measured concentrations are from Zhao et al. (2014). Here, both model predictions and measurements only include primary IVOCs. The predictions of IVOCs include all vapors in equilibrium with POA.**

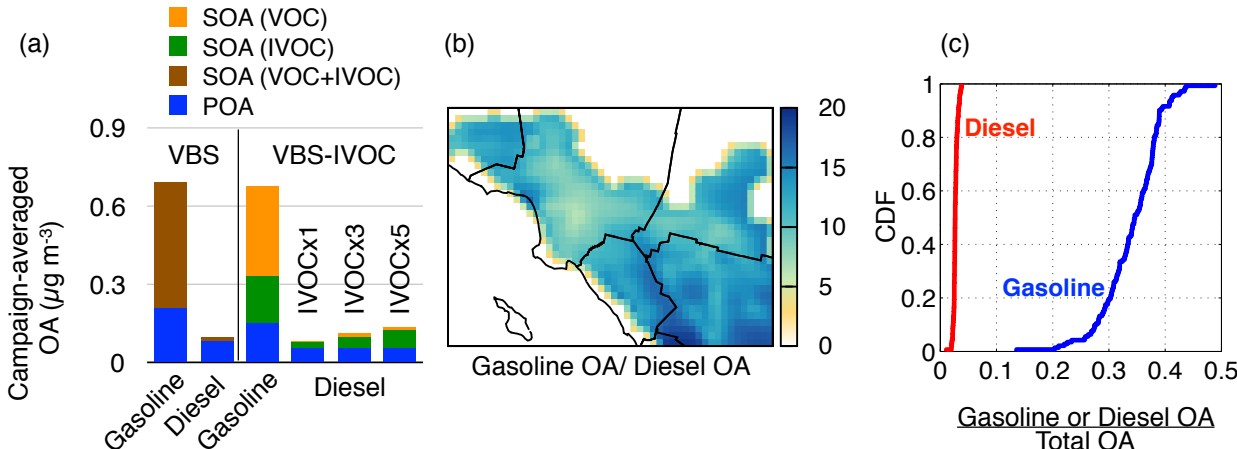

**Figure 6: (a) VBS-IVOC predicted campaign-averaged OA concentrations attributable to gasoline and diesel sources at the ground site in Pasadena; the IVOCx1 result for diesel use is from the VBS-IVOC simulation, the IVOCx3 and IVOCx5 results are from separate sensitivity simulations where IVOC emissions from diesel are scaled by a factor of 3 and 5 respectively as described in the text. (b) Ratio of gasoline OA to diesel OA over southern California and (c) cumulative distribution functions that show the fractional contribution of gasoline plus diesel OA to total OA in southern California.**