# Peer review of "Chemical Transport Model Simulations of Organic Aerosol in Southern California: Model Evaluation and Gasoline and Diesel Source Contributions"

_Atmospheric Chemistry and Physics, 2016_

## Referee Comment (RC1) · Anonymous Referee #1 · 30 Dec 2016

The manuscript by Jathar et al. summarizes results from a new modeling approach to characterize SOA formation in southern California. In this study, the volatility basis set along with updated VOC emission profiles and speciations are used in a chemical transport model to quantify SOA formation in the region and separate the contributions from diesel and gasoline sources. The topic is of high relevance given the interest in recent years in investigating SOA formation in urban areas from IVOCs and understanding the contribution of gasoline and diesel sources to SOA. The paper is well-written and organized, and the figures are of high quality. The only flaw in the structure of the manuscript is that the conclusion section is missing. I have a few technical and

minor editorial comments listed below. Once these concerns are addressed, I support publishing the manuscript.

Technical comments: P4, L22: Are the emission profiles of non-road gasoline sources assumed to be the same as on-road vehicles tested on the UC cycle? Also how are their emissions rates defined? Measurements in European cities in recent years have shown high amounts of SOA are formed from small (2- and 4-stroke gasoline engines). How are such emissions characterized for S. Cal?

P5, L 38-39: Why are biogenic SOA not aged similarly to other species?

P6, L2-3: It is mentioned that 10% of POA is shifted to SOA; is that because of POA aging or in addition to that?

Sec 3.1 and Fig 2: How are "episodes" defined? Also are these data limited to the boundary layer? Please clarify.

P6, L29: Isn't OM/OC ratio of 1.6 too low for SOA dominated regions and too high for POA dominated regions? Could uncertainties in this ratio also affect the comparison in Fig. 3a?

Fig 4. It is surprising the total POA from gasoline is more than that of diesel. Based on fuel-based emission factors of POA (Ban Weiss et al. 2008) and fuel use data, one expects the reverse. Are the emission factors used in this study significantly different than Ban Weiss et al.?

P8, L20: Since absolute amounts of the predicted IVOCs are in fact half of the measured values, why the emission ratios of IVOCs were not adjusted to match the measurements? Couldn't this also be the reason why the predicted OA is so much lower than the HR-AMS values in Pasadena (P7, L40)? Related to this, with the additional amounts of IVOCs in runs summarized in Figure 6a, how does the OA comparison with the Pasadena measurements look?

P11, L10-13: Is the low-NOx regime expected to be present downwind, i.e., in Inland

Empire or just over the ocean? Measurements over the land in S. Cal usually show NO levels at ∼100s of pptv which is much higher than the threshold for low-NOx conditions, given typical HO2 mixing ratios. Because of this, I don't think low-NOx conditions are common in S. Cal and therefore applying only the high-NOx SOA yields to VOCs/IVOCs should not change total SOA formation.

Minor comments: P3, line 9 and P5, L31 miss references P6, L28: Mention that HR-AMS measurements were made in Pasadena. Fig 3: Define in the caption what f.b. and f.e refer to. P7, L27-28: remove "in" from "note that in the VBS model...." and remove () around Woody et al. P7, L31-33: sentence is unclear and needs to be rephrased P9, L 19: add on: "...limited effect ON the SOA burden ".

―――――――――――――――――――

---

## Referee Comment (RC2) · Anonymous Referee #2 · 10 Jan 2017

This manuscript incorporated new experimentally derived inputs to improve the simulation of OA in souther California in the CMAQ model. The authors focused primarily on treatments of intermediate volatility organic compounds (IVOC) from gasoline and diesel sources, implementing updated estimates of emissions and updated parameterizations of SOA formation. They evaluated the simulated results against measurements during the CalNex campaign. Overall, the authors found the the updated model perfomed well at reproducing the (CSN) observed bulk OA concentrations at several locations in S. California. The updated model significantly underestimated OA concentrations when compared to HR-AMS measurements at Pasadena. However,

the updated model showed significant improvement at reproducing the OA composition and IVOC composistions at Pasadena. Model simulations showed that gasoline sources contribute about much more OA then diesel sources do due to the former's much larger SOA production. They showed that this conclusion is robust, even when the uncertainty in diesel IVOC emissions is considered.

In my view, this paper represent a good step in improving model representation of SOA formation under the VBS framework. Many current models use VBS, but the inputs to these models are highly uncertain, particularly for IVOC emissions and chemistry. I think the authors did a nice job at incorporating as much new experimentally-derived inputs as possible into their VBS model. The result is a updated, useful, and at least partially validated, model that the community can continue to build on.

I recommend publication after minor revision.

Comments:

Abstract: "The updated model, despite substatial differences in emissions and chemistry, performs similar to a recently released research version of CMAQ." This sentence is unclear. What is the "research version of CMAQ"? I assume it is the CMAQ used by Woody et al., without updated treatments to IVOC?

Figure 3: Are there CSN measurements at Pasadena that can be compared to the HR-AMS measurements? Also, why not compare model results to PMF ranalysis of the AMS measurement? I see this is done as a campaign average in Figure 4. But perhaps doing this comparison in Figure 3b would shed lights on why the high concentration days were more severely underestimated by the model. Might be worth a try.

Minor comments: There seems to be problems with the insertion of some citations (e.g., page 5, line 31 'ENREF_20').

Figure S3: The caption should read "Comparison of campaign-averaged predictions of the VBS model of Woody et al. (2016) and VBS-IVOC model".

[Figure]

---

## Author Response (AR1)

We thank both reviewers for their comments. We have revised the manuscript based on their comments and queries and provided a point-by-point response below. Reviewer comments are in *italic*, our response is in red and excerpts from the manuscript reflecting changes are in *italic blue*.

**Reviewer 1**

*The manuscript by Jathar et al. summarizes results from a new modeling approach to characterize SOA formation in southern California. In this study, the volatility basis set along with updated VOC emission profiles and speciations are used in a chemical transport model to quantify SOA formation in the region and separate the contributions from diesel and gasoline sources. The topic is of high relevance given the interest in recent years in investigating SOA formation in urban areas from IVOCs and understanding the contribution of gasoline and diesel sources to SOA. The paper is well- written and organized, and the figures are of high quality. The only flaw in the structure of the manuscript is that the conclusion section is missing. I have a few technical and minor editorial comments listed below. Once these concerns are addressed, I support publishing the manuscript.*

We agree with the reviewer's comment. To state the conclusions from the paper, we add a conclusion section to the manuscript:

"Conclusions:

[revised manuscript text omitted]

*2. P4, L22: Are the emission profiles of non-road gasoline sources assumed to be the same as on-road vehicles tested on the UC cycle? Also how are their emissions rates defined? Measurements in European cities in recent years have shown high amounts of SOA are formed from small (2- and 4-stroke gasoline engines). How are such emissions characterized for S. Cal?*

In this work, we only change the emissions profiles for mobile sources, not the magnitude of the emissions. The existing emissions inventory already accounts for the large differences in emissions rate between, for example 2-stroke and on-road gasoline sources. The NMOG speciation is assumed to be the same for on- and off-road gasoline sources. This assumption is supported by the results from May et al. (2014) who show that the NMOG speciation for 2-stroke off-road gasoline engines was consistent with that of on-road gasoline engines but found that the 4-stroke off-road gasoline engines produced more IVOC emissions. We should note that May et al. (2014) only quantify the NMOG speciation from a set of eight lawn and garden equipment engines, which may not be representative of the diversity found across the off-road sector. The on- and off-road equivalence in terms of the NMOG speciation is not a bad assumption but may need to be examined in detail by future work.

Emissions rates for off-road sources were calculated using MOVES (Baker et al., 2015). Platt et al. (2014) (the study the reviewer seems to be referring to) and Gordon et al. (2013) found that the production factors of SOA for 2- and 4-stroke gasoline engines were 2-3 orders of magnitude higher than those for light-duty vehicles. However, they showed that most of the SOA produced could be explained by the emissions and oxidation of aromatic compounds and did not find IVOCs to be an important precursor of SOA.

We have made a note about the points raised in this discussion in the section that talks about the gasoline and diesel contributions to OA: "*Platt et al. (2014) and Gordon et al. (2013) have recently argued that off-road sources, especially those powered using two-stroke engines, can be a large contributor to fine particle pollution in cities. In the inventory of Baker et al. (2015), which is used in this work, off-road sources contributed to ~40% of the NMOG and ~40% of the POA emissions from mobile sources. Given their substantial emissions, it is critical then that emissions rates from these sources be accurately represented in large-scale models. Only one study so far has reported VOC and IVOC emissions profiles from off-road engines. May et al. (2014) have found that two-stroke off-road gasoline engines had similar emissions profiles as on-road gasoline engines, but that the four-stroke off-road gasoline engines had much higher IVOC fractions than on-road gasoline engines. However, Platt et al. (2014) have shown that most of the SOA produced from two-stroke off-road gasolines engines can be explained by the emissions and oxidation of aromatic compounds and they did not find IVOCs to be an important precursor of SOA. In our work, we have assumed that the VOC speciation, IVOC fraction of NMOG, and the SOA parameterization for IVOCs were identical between the on- and off-road mobile sources. Given the uncertainties, these assumptions may need to be examined in detail in future work.* ".

*3. P5, L 38-39: Why are biogenic SOA not aged similarly to other species?*
The reason for not aging biogenic SOA is based on the modeling studies performed by Murphy and Pandis (2009) and Lane et al. (2008) where they observe that the aging of biogenic SOA results in the over-prediction of OA in the eastern United States. Additionally, Jathar et al. (2016) based on the extrapolation of chamber data suggest that aging reactions may double count SOA production and over-predict the importance of aging reactions in the atmosphere. The laboratory evidence for this is mixed. Donahue et al. (2012) show that OH oxidation of α-pinene SOA enhances SOA production while Henry and Donahue (2012) show that the OH oxidation combined with photolysis of the SOA products can result in destruction of SOA. We acknowledge this uncertainty for aging of biogenic (and anthropogenic) SOA in the manuscript: "*Murphy and Pandis (2009) report improved model performance when aging reactions are turned off for biogenic SOA. Recently, Jathar et al. (2016) proposed that laboratory chamber experiments that are used to parameterize SOA production may already include products from some aging reactions, raising concerns about double counting. Although some work has been done to understand the aging of biogenic SOA (Donahue et al., 2012;Henry and Donahue, 2012)}, future laboratory work needs to be directed in understanding the role of aging of OA vapors formed from anthropogenic sources on the mass and properties of OA.*".

*4. P6, L2-3: It is mentioned that 10% of POA is shifted to SOA; is that because of POA aging or in addition to that?*
The transfer of 10% of the POA mass to the SOA basis set is done as part of the aging reactions. The text is edited as follows: "*Finally, the aging reactions also shift a portion (~10%) of the POA vapors to the anthropogenic SOA basis set to maintain O:C ratios (Koo et al., 2014).*".

*5. Sec 3.1 and Fig 2: How are "episodes" defined? Also are these data limited to the boundary layer? Please clarify.*

The word 'episode' is used here to define the time period for which the model simulations are run. The time period simulated is chosen to coincide with the CalNex measurements. Since the word 'episode' is not required (and may create confusion), we have dropped it from the manuscript and the supporting information. For instance, the caption for Figure 1is edited to: "*Figure 1: Total emissions from May 4 to June 30, 2010 for POA, BTEX (aromatics), ALK5 (long alkanes) and IVOCs for gasoline and diesel sources in the Los Angeles and Orange Counties for the three OA models: Traditional, VBS and VBS-IVOC.*". The data presented in this work in Figures 1 through 6 are all limited to the surface/boundary layer to be consistent with the measurements.

*6. P6, L29: Isn't OM/OC ratio of 1.6 too low for SOA dominated regions and too high for POA dominated regions? Could uncertainties in this ratio also affect the comparison in Fig. 3a?*

We agree with the reviewer that the OM:OC ratio in the atmosphere spans a large range (~1.4-2.3) and depends on the source, composition, and photochemical age of the OA. This uncertainty obviously affects the model-measurement comparison in Figure 3(a). Our choice of a value of 1.6 is based on typical values used in the literature to convert filter-based OC measurements. This value is consistent with the average OM:OC ratio of 1.7 (with an uncertainty of ±30%) calculated by Hayes et al. (2013) from a detailed comparison of AMS based OA measurements with filter based OC measurements at Pasadena. We edit the manuscript to improve the discussion around the use of the 1.6 value: "*The CSN measurements need to be multiplied by an OM:OC ratio to account for the non-carbon species associated with organic carbon (Turpin and Lim, 2001). While ambient OM:OC ratio vary from 1.4 and 2.3 (Aiken et al., 2008), we use a constant value of 1.6 in this work based on previous estimates used for filter-based measurements (e.g., (Cappa et al., 2016)). This value is consistent with the OM:OC ratio of 1.7±0.5 estimated by Hayes et al. (2013) in Pasadena.*".

*7. Fig 4. It is surprising the total POA from gasoline is more than that of diesel. Based on fuel-based emission factors of POA (Ban Weiss et al. 2008) and fuel use data, one expects the reverse. Are the emission factors used in this study significantly different than Ban Weiss et al.?*

The reviewer is correct. If one uses the emission factor and POA:BC ratios for $PM_{2.5}$ from Ban-Weiss et al. (2008) for gasoline and diesel on-road sources (0.07 and 1.4 g kg-fuel$^{-1}$ and 0.71 and 2 respectively) and combines it with fuel use in the Los Angeles and Orange counties (46,000 tons day$^{-1}$ of gasoline and 5,300 tons day$^{-1}$ of diesel), we get POA emissions of 1.9 and 2.5 tons day$^{-1}$ for on-road gasoline and diesel sources respectively. In our work POA emissions, which are consistent with emissions from the EMFAC mobile source inventory for 2010, are 3.9 and 2.0 tons day$^{-1}$ for on-road gasoline and diesel sources respectively. This indicates that the on-road gasoline POA in our work is a factor of 2 higher than that estimated using the Ban-Weiss et al. (2008) data. If we use the findings from Ban-Weiss et al. (2008), the mobile POA fraction in Figure 4 will be reduced and deteriorate the comparison with the measured HOA fraction. However, the Ban-Weiss et al. (2008) data will not dramatically change the conclusions made from Figure 6 about the gasoline and diesel contributions to OA. We add the following sentence to address this comment: "*Comparison of the POA predictions from the VBS-IVOC model to ambient measurements made by Ban-Weiss et al. (2008) suggests that the on-road gasoline POA in the model may be over-predicted by a factor of 2, although this under-prediction does not significantly change the gasoline/diesel contribution to OA.*".

*8.P8, L20: Since absolute amounts of the predicted IVOCs are in fact half of the measured values, why the emission ratios of IVOCs were not adjusted to match the measurements? Couldn't this also be the reason why the predicted OA is so much lower than the HR-AMS values in Pasadena (P7, L40)? Related to this, with the additional amounts of IVOCs in runs summarized in Figure 6a, how does the OA comparison with the Pasadena measurements look?*

We thank the reviewer for this comment since it led us to refine and simplify our analysis. Our initial thought was that the spatial resolution of the model at 4 km probably dispersed the pollutants too much and was insufficient to simulate the absolute concentrations at the Pasadena ground site where the CalNex measurements were made. We therefore ratioed the IVOC measurements with CO for the model evaluation to account for the effects of dispersion. However, the correct quantity to compare, as others have used in the past (Woody et al., 2016), is the ratio of IVOC to ΔCO. However, as Woody et al. (2016) have pointed out, the ΔCO predictions are a factor of two to low compared to the ΔCO measurements and could mean that the CO emissions in the model might be a factor of two too low. We should note however, that the daily-averaged CO mixing ratios are only about 25% lower than the measurements at Pasadena. Since the intent here is to evaluate the IVOC concentrations and not evaluate the model's ability to predict CO (an under-prediction in CO does not necessarily suggest an under-prediction in non-methane organic gas (NMOG) emissions, a fraction of which – for combustion sources – is assumed to be IVOCs), we replot Figure 5 (see below) with absolute concentrations of IVOCs. While the total IVOC concentrations are under-predicted by ~35%, the new presentation of the data does not change the conclusions made in the manuscript. The text is edited based on the above-mentioned changes as follows: "*Gasoline and diesel IVOC concentrations (3.9 µg m$^{-3}$) from the VBS-IVOC model were 35% lower when compared to the hydrocarbon IVOCs concentrations measured by Zhao et al. (2014) (6 µg m$^{-3}$). In contrast predictions from the VBS model are a factor of 4 lower than the measurements, which highlights the improved representation of IVOCs in the VBS-IVOC model. The under-prediction of VBS-IVOC could partly be a result of the inability of the model with a 4 km horizontal resolution to capture the location-specific concentrations at Pasadena. The model-measurement comparison suggests that the VBS-IVOC model reasonably simulates the emissions, transport and chemistry of IVOCs from mobile sources. Furthermore, the VBS-IVOC model predicts that the majority of the hydrocarbon IVOCs originate from gasoline sources. Coincidentally, the predicted IVOC sum for other anthropogenic sources and biomass burning (4.3 µg m$^{-3}$) compared well with the measured oxygenated IVOCs (4.1 µg m$^{-3}$).*".

The additional amounts of IVOCs were only added for diesel-powered sources in Figure 6(a) to examine the sensitivity of IVOC emissions estimates on the gasoline-diesel OA split. Since all of these simulations only marginally change the diesel OA contribution, these would not have any effect on the model-measurement comparison in Figure 3b. This point is discussed in the Section *Discussion on Gasoline versus Diesel O*A: "*A factor of 5 increase in IVOC emissions only results in a 0.025 µg m$^{-3}$ increase in total OA mass concentration. Therefore, uncertainty in the diesel IVOC emissions does not appear to alter the model-measurement comparison discussed earlier.*".

[Figure]

*Figure 5: Comparison of predicted and measured campaign-averaged IVOC concentrations at the Pasadena ground site. Measured concentrations are from Zhao et al. (2014). Here, both model predictions and measurements only include primary IVOCs. The predictions of IVOCs also include primary vapors in equilibrium with POA.*

*9. P11, L10-13: Is the low-NOx regime expected to be present downwind, i.e., in Inland Empire or just over the ocean? Measurements over the land in S. Cal usually show NO levels at ~100s of pptv which is much higher than the threshold for low-NOx conditions, given typical HO2 mixing ratios. Because of this, I don't think low-NOx conditions are common in S. Cal and therefore applying only the high-NOx SOA yields to VOCs/IVOCs should not change total SOA formation.*

We agree with the reviewer that when considering the precursor contributions at Pasadena, the NO$_x$ dependence probably does not help explain differences between the 3D model predictions from this work and the box model predictions from (Jathar et al., 2014). This is because the NO levels are probably high enough that most of the SOA formation proceeds through the high NOx pathway. We thank the reviewer for this comment and we edit the text to only consider the effect of aging to explain the difference in the 3D and box model results: "*Figure 6(a) resolves the OA contributions based on the precursor class at the Pasadena site. The VBS-IVOC model predicts that IVOCs, particularly from gasoline vehicles, form almost as much SOA as VOCs (long alkanes and single-ring aromatics). This is in contrast to Jathar et al. (2014), who found that unspeciated precursors (or IVOCs) were approximately a factor of 4 larger than VOCs in forming SOA in chamber experiments. One possible explanation for this difference is that Jathar et al. (2014) did not account for the effects of continued aging of IVOC oxidation products on OA concentrations*".

*10. Minor comments: P3, line 9 and P5, L31 miss references*
Those references were accidently added by EndNote. They have been removed.

*P6, L28: Mention that HR- AMS measurements were made in Pasadena.*
We modify the sentence to mention Pasadena: "*Figure 3(b) compares predictions of daily-averaged OA concentrations to measurements made using a high-resolution aerosol mass spectrometer (HR-AMS) in Pasadena (Hayes et al., 2013)*".

*Fig 3: Define in the caption what f.b. and f.e refer to.*

The following text is added to the caption: "*f.b. is the fractional bias ($\frac{1}{N}\sum_{i=1}^{N}\frac{P-M}{\frac{P+M}{2}}$) and f.e. is the fractional error ($\frac{1}{N}\sum_{i=1}^{N}\frac{|P-M|}{\frac{P+M}{2}}$); P is the predicted value, M is the measured value and N is the sample size.*".

11. P7, L27-28: remove "in" from "note that in the VBS model. . .." and remove () around Woody et al.
The text is corrected.

12. P7, L31-33: sentence is unclear and needs to be rephrased
We revise the sentence to: "*It is unclear if the predicted non-mobile, non-cooking and non-biomass burning POA (which in Pasadena accounts of ~9% of the OA) should be added to the SOA predictions before being compared with the OOA factor derived from the ambient tdata. The non-mobile, non-cooking and non-biomass burning POA (or anthropogenic (other) POA) category here includes sources such as stationary fuel combustion (e.g., natural gas combustion), surface coatings (e.g., metal coating), mineral processes (e.g., concrete production), road dust and managed burning (e.g., prescribed burns).*".

13. P9, L 19: add on: ". . .limited effect ON the SOA burden ".
The text is corrected.

**Reviewer 2**

*This manuscript incorporated new experimentally derived inputs to improve the simulation of OA in southern California in the CMAQ model. The authors focused primarily on treatments of intermediate volatility organic compounds (IVOC) from gasoline and diesel sources, implementing updated estimates of emissions and updated parameterizations of SOA formation. They evaluated the simulated results against measurements during the CalNex campaign. Overall, the authors found the updated model performed well at reproducing the (CSN) observed bulk OA concentrations at several locations in S. California. The updated model significantly underestimated OA concentrations when compared to HR-AMS measurements at Pasadena. However, the updated model showed significant improvement at reproducing the OA composition and IVOC compositions at Pasadena. Model simulations showed that gasoline sources contribute about much more OA then diesel sources do due to the former's much larger SOA production. They showed that this conclusion is robust, even when the uncertainty in diesel IVOC emissions is considered. In my view, this paper represents a good step in improving model representation of SOA formation under the VBS framework. Many current models use VBS, but the inputs to these models are highly uncertain, particularly for IVOC emissions and chemistry. I think the authors did a nice job at incorporating as much new experimentally-derived inputs as possible into their VBS model. The result is an updated, useful, and at least partially validated, model that the community can continue to build on. I recommend publication after minor revision.*

1. Abstract: "The updated model, despite substantial differences in emissions and chemistry, performs similar to a recently released research version of CMAQ." This sentence is unclear. What is the "research version of CMAQ"? I assume it is the CMAQ used by Woody et al., without updated treatments to IVOC?
Yes, the reviewer is correct and we edit the abstract to be more clear: "*The updated model, despite substantial differences in emissions and chemistry, performed similar to a recently*

*released research version of CMAQ (Woody et al., 2016) that did not include the updated VOC and IVOC emissions and SOA data.*".

*2. Figure 3: Are there CSN measurements at Pasadena that can be compared to the HR- AMS measurements? Also, why not compare model results to PMF analysis of the AMS measurement? I see this is done as a campaign average in Figure 4. But perhaps doing this comparison in Figure 3b would shed lights on why the high concentration days were more severely underestimated by the model. Might be worth a try.*

Unfortunately, there are no CSN sites in or near Pasadena. However, there were co-located filter measurements performed by research groups other than those using the HR-AMS at Pasadena. Hayes et al. (2013) perform a comprehensive comparison of the HR-AMS data against the filter measurements (described in the supporting information) and find that the HR-AMS data were generally consistent with the filter measurements and there was no indication that the HR-AMS data are over-estimating OA mass concentrations. Woody et al. (2016) compare hourly-averaged model predictions from CMAQ against PMF factors from the HR-AMS data and make the following conclusions: (i) cooking OA concentrations compare well with AMS-COA during the morning but are under-predicted in the afternoon and late night, (ii) non-cooking POA concentrations compare well with AMS-HOA but are under-predicted during the afternoon, and (iii) predicted SOA concentrations capture diurnal trends in OOA but are consistently a factor of 5 lower during all times of the day. Since the model predictions of OA mass concentrations and diurnal profiles in this work did not change dramatically when compared to Woody et al. (2016), the findings described earlier apply here. We add a short discussion about this in the text: "*Before discussing the normalized composition predicted by the VBS-IVOC model, we briefly describe the findings from Woody et al. (2016) who carefully compared the predictions of absolute concentrations of the VBS model to the PMF factors estimated from the ambient HR-AMS measurements. Woody et al. (2016) found that (i) the predicted cooking-related OA concentrations compared well with the COA factor during the morning but were low in the afternoon and late night, (ii) non-cooking POA concentrations compared well with the HOA factor except during the afternoon when it was underpredicted, and (iii) predicted SOA concentrations match the diurnal profile of the OOA factor but were but was a factor of 5 lower during all times of the day.*".

*3. Minor comments: There seems to be problems with the insertion of some citations (e.g., page 5, line 31 'ENREF_20').*

The citations is fixed in the revised manuscript.

*4. Figure S3: The caption should read "Comparison of campaign-averaged predictions of the VBS 
[revised manuscript text omitted]

When compared to a Traditional model of OA in CMAQ that includes a non-volatile treatment of POA and no SOA from IVOCs, the VBS-IVOC model produced different spatial patterns of OA with lower (~50%) concentrations in source regions but higher (~20-40%) concentrations away from the sources. In comparison to the Traditional model, the VBS-IVOC model better predicted the sources and composition of OA. These findings are consistent with previous comparisons between Traditional- and VBS- models and highlight the importance of the use of an OA model that includes semi-volatile and reactive POA and SOA formation from IVOCs.

Predictions of OA from the VBS-IVOC model are similar to those from a recently released research version of CMAQ (VBS) that included semi-volatile POA and SOA formation from IVOCs (Woody et al., 2016). The predictions of these two models were similar for three reasons. First, the VOC and IVOC updates in this work, surprisingly, did not substantially alter the total emissions of SOA precursors in southern California (although the VOC-IVOC composition was different between the two models for gasoline sources). Second, mobile sources only accounted for slightly more than one-third of the total OA in southern California and hence updates to the emissions and SOA production from mobile sources had a limited influence on the total OA burden. And third, and most important, is that both models predict that multigenerational aging of vapors in equilibrium with OA is a major source of SOA. In addition, both models used similar aging mechanisms that are 
[revised manuscript text omitted]

**Comment [ALR3]:** Different y-axis scales. Do you want to change max of diesel to 300 so that it is a simple multiple of gasoline?

[Figure]

**Figure 2: Averaged predictions from the VBS-IVOC model for (a) total OA (µg m$^{-3}$), (b) POA fraction, (c) SOA fraction, (d) total gasoline OA (µg m$^{-3}$), (e) total diesel OA (µg m$^{-3}$), (f) biogenic SOA (µg m$^{-3}$) and (g) other OA (µg m$^{-3}$) over southern California.**

[Figure]

[Figure]

**Figure 3: Scatter plot of VBS-IVOC OA predictions versus** 24-hr measurements from (a) filters collected at sites in the Chemical **Speciation Network (CSN) and (b)** HR-AMS measurements at the **Pasadena ground site during the CalNex campaign. In panel (a) the model-measurement comparison is for six sites in California (Fresno, Bakersfield, Central Los Angeles, Riverside, El Cajon and Simi Valley).** f.b. is the fractional bias ($\frac{1}{N}\sum_{i=1}^{N}\frac{P-M}{\frac{P+M}{2}}$) and f.e. is the fractional error ($\frac{1}{N}\sum_{i=1}^{N}\frac{|P-M|}{\frac{P+M}{2}}$); $P$ is the predicted value, $M$ is the measured value and $N$ is the sample size.

[Figure]

**Figure 4:** Averaged, **normalized composition of OA at the Pasadena ground site as predicted by the Traditional and VBS-IVOC models. Predictions are compared to PMF factors** derived from ambient **HR-AMS data** collected in Pasadena **Hayes et al. (2013)**.

[Figure]

Figure 5: Comparison of predicted and measured average IVOC concentrations at the Pasadena ground site. Measured concentrations are from Zhao et al. (2014). The predicted IVOCs include primary vapors in equilibrium with POA. The data have been normalized by carbon monoxide (CO) concentrations to correct for any differences in mixing.

**Comment [ALR4]:** ?  All varpors or just the ones you classified as IVOCs?

**Comment [ALR5]:** Delta?

[Figure]

Figure 6: (a) VBS-IVOC predicted campaign-averaged OA concentrations attributable to gasoline and diesel sources at the ground site in Pasadena; the IVOCx1 result for diesel use is from the VBS-IVOC simulation, the IVOCx3 and IVOCx5 results are from separate sensitivity simulations where IVOC emissions from diesel are scaled by a factor of 3 and 5 respectively as described in the text. (b) Ratio of gasoline OA to diesel OA over southern California and (c) cumulative distribution functions that show the fractional contribution of gasoline plus diesel OA to total OA in southern California.

reason could be attributed to the representation of aging reactions in the models, which

The aging reactions simulated the multigenerational gas-phase oxidation of vapors in equilibrium with OA. The aging mechanism used here iwas conceptually based on the work of Robinson et al. (2007), which assumed a constant reaction rate constant and only allowsed for the formation of functionalized, lower-volatility products. In essence, aging reactions, with enough time, will convert all semi-volatile vapors into particles. However, the mechanism assumes a constant reaction rate constant,However, reaction rates which may vary with C* and O:C of the OA and does not account for fragmentation reactions, which should become can be increasingly important at longer time scales (Kroll et al., 2011). The reaction rate for aging and the effects of aging at longer time scales Neither of these have not been constrained against laboratory data. This implies that the OA predictions, despite the substantial new data, becomeare poorly constrained as one moves downwind of source regions. Murphy and Pandis (2009) have found that model predictions agree better with measurements when aging reactions are turned off for biogenic SOA. Recently, Jathar et al. (2016) have shown that aging reactions similar to those suggested in Robinson et al. (2007) might not be necessary since the laboratory chamber experiments that are used to parameterize SOA production already include products from the aging reactions happening inside the chamber. The work of Murphy and Pandis (2009) (for biogenics) and Jathar et al. (2016) suggests that including aging reactions in CTMs may double count SOA production and over-predict the importance of multigenerational gas-phase chemistry in the atmosphere. Although some work has been done to understand the aging of biogenic SOA (Donahue et al., 2012;Henry and Donahue, 2012), future laboratory work needs to be directed in understanding the role of aging of OA vapors formed from anthropogenic sources on the mass and properties of OA.

---

## Author Response (AR2)

We thank the editor for her comments and have revised the manuscript accordingly.

1. Page 2 lines 32-33 "for example, inclusion of semi-volatile POA without SOA formation from IVOCs and aging reactions degraded model performance vis-à-vis total OA mass." Please include the reference for this result.
We have added the reference Robinson et al. (2007).

2. Page 8 line 1: "Biogenic SOA was not aged." Although the reason is included in the conclusions, it would perhaps be better to mention that here as well.
We have revised the sentence to offer justification for why biogenic SOA was not aged.

3. Page 13 line 32-pg 14 line 3 "Comparison of the POA predictions from the VBS-IVOC model to ambient measurements made by Ban-Weiss et al. (2008) suggests that the on-road gasoline POA in the model may be over-predicted by a factor of 2, although this under-prediction did not significantly change the gasoline/diesel contribution to OA." This sentence is very unclear to me. The POA is too high in the model but then the under-prediction is discussed. Should this be over-prediction?
Yes, it should have read over-prediction. We have corrected this in the revised manuscript.

4. Pg 15 line 19: This should reference Section 3.4
Corrected.

5. Page 17 line 21: Is 13 the domain average? On page 13 line 26-27 a range for the gasoline contribution to OA of 10-20 times diesel is given. Please clarify.
The 13 value was the median. We have revised the sentence on page 13 line 26-27 to be consistent with the 13 value mentioned in the abstract and conclusions.

Supplement
ALK5 is not included in table S.2 despite its reference in the caption and in the main text. Please update the table.
We have added emissions of ALK5 in Table S.2.